# Efficient Microwave Filter Design by a Surrogate-Model-Assisted Decomposition-Based Multi-Objective Evolutionary Algorithm

**Yongfeng Wei [1], Guangfei Qi [1], Yanxing Wang [1], Ningchaoran Yan [1], Yongliang Zhang [2],\* and Linping Feng [3,4]**

[1] School of Electronics and Information Engineering, Inner Mongolia University, Hohhot 010021, China
[2] School of Transportation Institute, Inner Mongolia University, Hohhot 010021, China
[3] Shaanxi Key Laboratory of Deep Space Exploration Intelligent Information Technology, School of Electronics and Information Engineering, Xian Jiaotong University, Xi'an 710049, China
[4] School of Microelectronics, Xian Jiaotong University, Xi'an 710049, China
\* Correspondence: namar@imu.edu.cn; Tel.: +181-4834-2159

**Abstract:** As a crucial frequency selection device in modern communication systems, the microwave filter plays an increasingly prominent role. There is a great demand for the multi-objective design of microwave filters. The filter's performance affects the quality of the whole communication system directly. However, traditional multi-objective electromagnetic (EM) optimization design demands repetitive EM simulations to adjust the physical parameters of the microwave filters. Accordingly, using electromagnetic simulation directly to design optimization is quite expensive. Given this situation, this paper applies a novel surrogate model based on one-dimensional convolutional autoencoders (1D-CAE) into the multi-objective algorithm evolutionarily based on decomposition (MOEA/D) for the first time. This approach uses MOEA/D as the multi-objective optimizer, and a novel low-complexity surrogate model based on one-dimensional convolutional autoencoders (1D-CAE) is constructed to predict the expensive EM simulation results. The surrogate model based on 1D-CAE is used to generate the results of scalar subproblems of MOEA/D, which greatly improves the design efficiency. Compared with the traditional design methods based on an EM solver, this method not only effectively optimizes multiple design objectives but also completes the design of microwave filters in a shorter time. The proposed method is verified using the design of a sixth-order ceramic filter and a seventh-order metal cavity filter.

**Keywords:** multi-objective optimization; microwave filter design; MOEA/D; surrogate model; one-dimensional convolutional autoencoders

## 1. Introduction

The microwave filter is quite a vital RF device in wireless communication systems. With the rapid development of wireless communication technology, it is very important to improve the efficiency of filter design. In recent years, a variety of optimization algorithms, such as particle swarm optimization (PSO), ant colony optimization (ACO), and differential evolution (DE), are receiving widespread attention. PSO, ACO, DE, and population-based incremental learning (PBIL) have gained recognition in filter design [1–6]. However, in the design process of microwave filters, engineers cannot just consider a single design objective but need to achieve equilibrium among multiple different objectives, so as to make microwave filters meet the prescribed requirement. For example, in the wideband balun bandpass filters (BPFs), the phase imbalance, magnitude imbalance, and frequency responses are important indicators to determine the filter performance. If any response parameter fails to meet the prescribed requirement, it will directly affect the final performance of the filter balun structure [7]. Therefore, multi-objective optimization is gradually replacing single-objective optimization and has become the mainstream optimization form of modern microwave filter design [8–10].

For the multi-objective optimization method, MOEA/D [11] is one of the most commonly used methods for multi-objective optimization problems. MOEA/D uses the aggregation function to disassemble multi-objective problems into several scalar optimization subproblems with neighborhood relations and to co-evolve rather than relying on Pareto domination. Due to the decomposition operation, MOEA/D has prominent advantages to maintain the distribution of solutions. Several MOEA/D variants have been proposed to improve MOEA/D. A variant called MOEA/D-ACDP is proposed in [12], which improves the population diversity in infeasible regions through the angular information of the objective function. Qi et al. proposed a variant that can adaptively adjust weights (MOEA/D-AWA) [13], which initializes and adjusts the weight vector in a new way. A form of fusion (MOEA/DD) that combines decomposition and domination is proposed in [14]. Li et al. proposed MOEA/D-STM [15], which utilizes a matching model to coordinate the selection process in MOEA/D to balance convergence and diversity.

In addition to a better optimization algorithm, a more important problem is how to improve the efficiency of multi-objective microwave filter design. Recently, EM-based optimization methods were developed into a vital optimization approach for microwave filter design. The meshes applied to EM simulations are usually produced by mesh adaptation methods. For traditional coarse- and fine-mesh SM, as long as the geometric parameter value changes, it is necessary to regenerate the coarse mesh to adapt to this change. However, in the optimization design of microwave filters, a great deal of EM simulations is required to acquire the optimal design space parameters commonly. Therefore, the traditional direct EM-based microwave filter design takes a long time to meet design requirements. Space mapping (SM) is the key technology to solving the above problem. SM has shown great usability in computer-aided optimization design [16–19]. The SM concept combines the computational efficiency of coarse models with the accuracy of fine models [20]. Although the fine models are accurate, they may be expensive. The SM technique constructs a mathematical relation between the coarse and the fine models, orients a large number of CPU-intensive computations to the coarse model, and preserves the accuracy offered by the fine model [21]. At present, the maturity of applying SM to microwave component optimization design is increasing day by day [22–25].

In recent years, in order to improve the efficiency of filter design, many optimization methods based on SM have been proposed. Still, these methods only consider specific types of filter structures [26]. A surrogate-model-assisted PSO algorithm is proposed in [27]. The algorithm can effectively shorten the design time, but it can only be used for single-objective microwave filter design. Hence, most of the available methods cannot combine better optimizers and shorter design time at the same time. In [28], a novel off-line surrogate model is proposed to design filters with more than 10 variables. More than 2000 samples are used to construct this single high-accuracy surrogate model [28]. However, this is unrealistic for more complex filters, and most filter design/optimization methods try to avoid such a step.

To address the above problems, a novel microwave filter design technique is presented, called MOEA/D based on 1D-CAE. Convolutional autoencoder (CAE) is an important type of deep learning model that is widely applied into various fields, such as image denoising, neural style transfer, and so on. Masci et al. [29] developed CAE to process 3D image data. In [30], a deep CAE is proposed to process high-resolution SAR images. Combining stack and CAE, [31] put forward a new type of structure of low-light image enhancement. In [32], a three-layer CAE architecture and an effective algorithm are designed to learn CAE parameters for device-free localization. However, CAE, as a surrogate model, is introduced into the research of microwave filter design, which is still blank. Only [27] has made efforts in this regard, but it cannot complete the design of a multi-objective microwave filter. Therefore, in this paper, CAE and MOEA/D are combined for the first time to realize the fast multi-objective design of microwave filters.

In this paper, the 1D-CAE network structure predicts the filter's S-parameters at potentially better sampling points by building an inexpensive surrogate model. Autoencoder is

an artificial neural network that learns the efficient representation of input data. However, in the traditional autoencoder training process, it is very easy to copy the input to the output, which leads to the poor predictive ability of the trained model. The model's prediction ability will be greatly reduced when the input samples are too complex. For filter design, the relationship between the physical parameters of the filter and the S-parameters is complex. Therefore, it is difficult for autoencoders to extract effective features from filter data. The convolutional neural network (CNN) can perceive adjacent dimensions of data and obtain local features through receptive fields and parameter sharing. This process is based on convolution of the kernel. A variety of local features can be obtained by constructing data samples with multiple convolution kernels. In addition, the pooling layer can reduce the parameters in the whole neural network. Therefore, iterative convolution and pooling can be utilized to extract the final features of the data samples. The framework of 1D-CAE is as follows. In the encoder part, features are extracted by CNN, which continuously iterates the convolution and pooling of multiple convolution kernels to reduce the number of features. In the decoder part, the extracted features are used to reconstruct the sample data by the reshape operation and full connection layer. It can be said that CAE combines the advantages of CNN and AE in feature learning and data reconstruction.

In this paper, the 1D-CAE network structure uses the one-dimensional convolutional neural network (1D-CNN) as an encoder for feature extraction. That is, the relationship between a microwave filter's physical parameters and responses is established. After fully learning by the encoder, the physical parameters are predicted into the characteristic response by the decoder directly. As shown in Figure 1, the decoding part of 1D-CAE is applied to construct a novel surrogate model to displace the full-wave EM simulation, which greatly shortens the calculation time of MOEA/D. This paper uses the loss function to determine whether the 1D-CAE prediction is accurate. The MOEA/D algorithm based on the 1D-CAE network structure aims to:

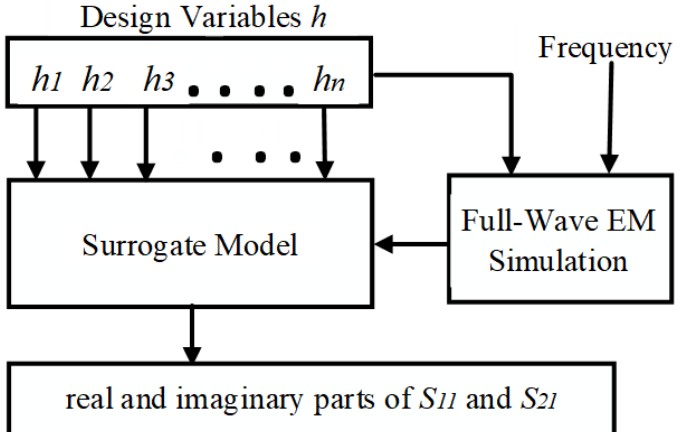

**Figure 1.** Surrogate model constructed by 1D-CAE.

(1) Multiple design objectives can be jointly optimized;

(2) Obtain comparable results as the traditional multi-objective design approach (directly using EM solver for MOEA/D);

(3) Finish the multi-objective design of the filter in a shorter time (a few hours to one day or so);

(4) When the specific characteristics of the filter are not considered, it can be universally used in most types of filters.

The remainder of this article is structured as follows. Section 2 discusses related works and methods, illustrates the basic theory of 1D-CAE network structure and the MOEA/D algorithm in terms of mathematical formulation, and introduces a detailed framework of 1D-CAE-based MOEA/D. In Section 3, the practicability and effectiveness of the presented

technology are confirmed by two practical microwave filter design examples. Section 4 concludes the paper.

## 2. Materials and Methods

### 2.1. D-CAE Network Structure

2.1.1. Autoencoder's Framework

Autoencoder is an artificial neural network that learns the efficient representation of input data. It is trained to try to reproduce input to output. The dimension of the encoder part is universally much smaller than that of the input data. So, AE can be applied for dimensionality reduction. By extracting features, AE also has the function of generating model, that is, it can randomly generate data similar to training data.

As shown in Figure 2, the typical structure of AE has two parts (encoder part and decoder part). The encoder part is composed of an input layer and a hidden layer, and it is used to extract latent features from original inputs. The original input data are $X = (x_1, \ldots, x_n)$, and the function of an encoder is to map the input sample into the latent space as the feature $h$:

$$h = Encoder(X), \tag{1}$$

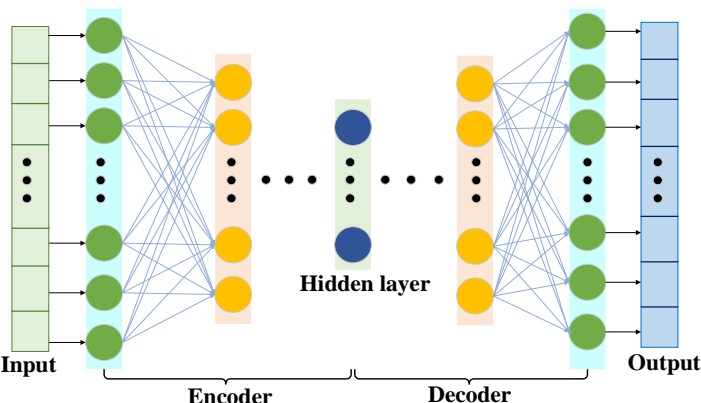

**Figure 2.** The typical structure of AE.

The decoder part is composed of a hidden layer and an output layer. To obtain a decoded output vector $X*$, the decoder part maps the feature $h$ to the original input space:

$$X* = Decoder(h), \tag{2}$$

AE is committed to infinitely narrowing the gap between input samples and output samples. This gap is represented by the loss function:

$$L = K(X, X*), \tag{3}$$

2.1.2. One-Dimensional Convolutional Autoencoders

The traditional autoencoder can easily copy the input sample to the output sample. However, the scope of application of this model is limited. The model's prediction ability will be greatly reduced when the input samples are too complex. For filter design, the relationship between the physical parameters of the filter and the *S*-parameters is complex. In this paper, the 1D-CAE network structure based on AE and 1D-CNN is adopted to build the surrogate model. Including three convolution layers, three pooling layers, a reshape module, and a full connection layer, namely conv1, pooling1, conv2, pooling2, conv3, pooling3, reshape1, and full-con1, the encoder module is used to encode the input data. The decoder module consists of the reshape operation and full connection layer, namely full-con2, full-con3, full-con4, and reshape2. Figure 3 is an internal detailed structure diagram of 1D-CAE. Different types of layers are described in detail below.

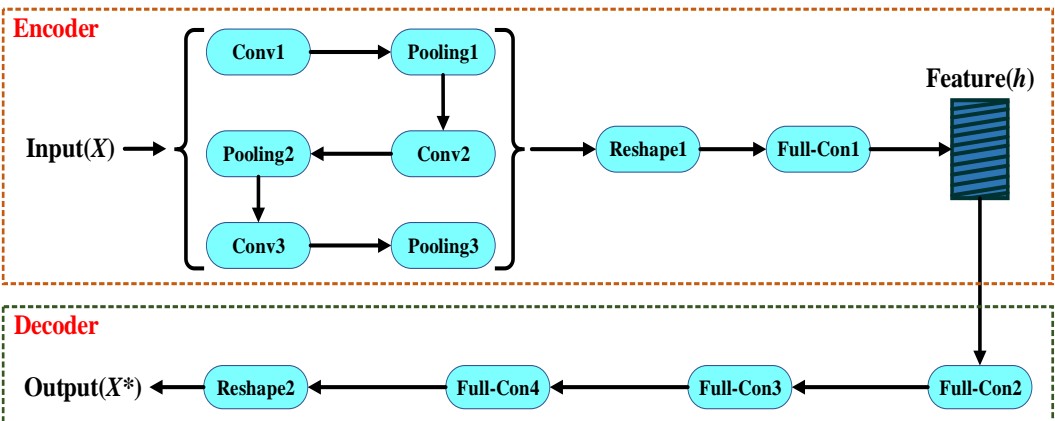

**Figure 3.** The detailed structure of the 1D-CAE network structure.

As a feature extractor, the convolution layer makes use of several convolution kernels for convolution calculation to obtain more features. Let $f_z$ indicate the output of the *Z-th* kernel. $f_z$ is determined as follows [33]:

$$f_z = \text{ReLU}\left(\sum X \odot \omega_z + b_z\right), \tag{4}$$

$$\text{ReLU}(x) = \max(0, x), \tag{5}$$

where $\odot$ represents convolution operation, $\omega_z$ is the *Z-th* convolution kernel, and $b_z$ is the offset corresponding to the *Z-th* convolution kernel. *Z* indicates the number of channels. ReLU is a rectifier linear unit.

The pooling layer can reduce the parameters in the whole neural network. The common types of pooling layers are max pooling and average pooling. This paper adopts max pooling. Max pooling only retains the most significant features so that it can reduce the size of the training sample and improve computational efficiency. For the *k*-length feature of the convolution layer in the *Z-th* channel, the output of the pooling layer is determined by the following formula [33]:

$$P_z(n) = \max_{0 \le n \le \frac{k}{st}} \{f_z(nS, (n+1)S)\}, \tag{6}$$

where $f_z$ represents the input. *S* is the size of the pooling window, and *st* is the step size.

The reshape module transforms the extracted feature vector into a one-dimensional form to facilitate coding in the full connection layer. The calculation process is as follows [34]:

$$F = \sin\left(w_f * E + b_f\right), \tag{7}$$

where *E* is the eigenvector output by the last level pooling layer, $w_f$ is the weight of the full connection layer, $b_f$ indicates the offset of a full connection layer, and sin represents a kind of activation function.

### 2.1.3. 1D-CAE Network Structure Introduction in Filter Design

For the encoding part of the 1D-CAE, the input is the real part and imaginary part of the filters *S11* and *S21*, and the output is the geometric parameters of the filter to be optimized. On the contrary, the input of the decoding part is the geometric parameters to be optimized, and the real and imaginary parts of *S11* and *S21* are their outputs. Therefore, the whole 1D-CAE network structure is a system in which the input part is the real *S11* and *S21* and the output part is the predicted *S11* and *S21*, that is, the generation model function of the 1D-CAE network structure.

For the actual optimization of this paper, the performance of the 1D-CAE network structure is reflected by the combination of rebuild loss ($L_r$) and forecast loss ($L_f$). Let us

use rebuild loss to assess the performance of 1D-CAE network structure reconstruction filter features. Rebuild loss is determined by the following formula [27]:

$$L_r = \frac{1}{M} \sum_{i=1}^{M} |S_{em} - S_{pr}|,\tag{8}$$

where $M$ represents the total training data, $S_{em}$ represents the simulated $S$-parameters, and $S_{pr}$ is the $S$-parameters of the reconstructed 1D-CAE network structure.

The predicted losses are expressed as follows [27]:

$$L_f = \frac{1}{M} \sum_{i=1}^{M} |H_{em} - H_{pr}|,\tag{9}$$

where $M$ represents the total training data, $H_{em}$ represents the realistic value of the physical parameters, and $H_{pr}$ is the predicted physical parameters' value.

The total loss function is determined by the following formula [27]:

$$L_t = L_r + kL_f,\tag{10}$$

where $k$ is a regularization parameter. So as to make the network prediction more accurate, it is essential to minimize the value of the loss function ($L_t$). In this paper, the Adam optimizer is used to optimize the loss function of the network.

### 2.2. MOEA/D Algorithm
2.2.1. Multi-Objective Optimization Problem

The multi-objective optimization problem (MOPs) makes multiple objectives as best as possible in a given region at the same time. It is expressed by the following formula:

$$\text{Minimize} f(x) = (f_1(x), \dots, f_M(x)) \quad \text{subject to } x \in \text{X},\tag{11}$$

where $x = (x_1, \dots, x_e)$ represents the e-dimensional vector to be solved, X represents the boundary range of $x$, $f_i(x)$ represents the $i$-th objective function to be optimized, and $M$ is the dimension of the goal vector. MOPs cannot receive a single solution that can meet all objectives concurrently. Accordingly, the solution of MOPs is usually a set of equilibrium solutions. Suppose $x^E$ and $x^F$ are two solutions of the MOPs in (11); $x^E$ is said to Pareto dominate $x^F$ if and only if $f_i(x^E) \leq f_i(x^F)$ for all $i \in \{1, 2, \dots, M\}$ and $f_j(x^E) \leq f_j(x^F)$ for at least one $j \in \{1, 2, \dots, M\}$. A solution $x^E$ is Pareto optimal when there is no $x^G \in \text{X}$ that Pareto dominates $x^E$. The Pareto optimal set consists of all Pareto optimal solutions, and the Pareto front (PF) is the projection of the Pareto optimal set in the objective space.

2.2.2. MOEA/D

MOEA/D disassembles the MOPs into several single-objective optimization subproblems with neighborhood relations, and each subproblem is a dissimilar set of all objectives. Then, by analyzing the information of adjacent problems, all subproblems are optimized concurrently by an evolutionary algorithm. Due to the decomposition operation, this method can well maintain the distribution of solutions. In addition, by analyzing the information of adjacent problems to optimize, it is effective to avoid falling into local optimum. The detailed introduction to the important part of the MOEA/D is as follows.

I.    Weight vector

In order to disassemble the MOPs into several single-objective subproblems, MOEA/D needs to distribute even weight vectors in the target space. The number of weight vectors is the same as the population size. If the population size is $N$, the number of weight vectors is $N$. Each weight vector turns the MOPs into a single-objective problem. $N$ sets of weight

vectors are $N$ single-objective optimization problems and $w_N = (w_{N1}, w_{N2}, \ldots, w_{NM})$ is the *N-th* weight vector, so:

$$w_{N1} + w_{N2} + \ldots + w_{NM} = 1, \tag{12}$$

where $M$ represents the number of problems to be optimized. In addition, the uniformity of weight vector distribution should be as good as possible to improve the accuracy of optimization results.

II.    Decomposition strategy

The MOEA/D aims to convert the multi-objective optimization problem to single-objective optimization problem through an aggregation function. The generally applied decomposition strategies include penalty-based boundary intersection (PBI), weighted sum (WS), and Tchebycheff (TCH), among which the WS cannot solve nonconvex functions. The Tchebycheff method is used in this paper. The formula of Tchebycheff method is expressed by the following formula:

$$\text{Minimize } g^{\text{TCH}}(x \,|\, w, z^*) = \max_{i=1,2,\ldots,M} \{w_i \cdot |z_i^* - f_i(x)|\}, \tag{13}$$

where $z$ is the optimization result to be achieved (the reference point). Take double objectives as an example, calculating $w_1 \cdot |z_1^* - f_1(x)|$ and $w_2 \cdot |z_2^* - f_2(x)|$, respectively, to take the maximum value. The larger the value, the farther away from the reference point on this objective function. Supposing $w_1 \cdot |z_1^* - f_1(x)|$ is larger, please gradually change $x$ so that this value is closer and closer to $z^*$ until it reaches the corresponding point on Pareto front. This process is actually finding the minimum value of the function $g^{\text{TCH}}(x) = w_1 \cdot |z_1^* - f_1(x)|$. If $w_1 \cdot |z_1^* - f_1(x)|$ reaches its minimum value, $w_2 \cdot |z_2^* - f_2(x)|$ will also reach its minimum value. This is true for weight vector $w$. Each weight vector obtains the corresponding solution in this way.

III.   Neighborhood structures

Neighborhood structure is determined by the Euclidean distance among the weight vectors, which is the key to generating a new solution. The MOEA/D assumes that the solutions on adjacent weight vectors are similar, and each weight vector has neighbors. After generating a new solution, compare the newly generated solution with all solutions in its neighborhood (including the current solution of the current subproblem). When the newly generated solution is better, all poor neighbors will be displaced by the newly generated solution.

### 2.3. Filter Design by Surrogate-Modeling-Assisted MOEA/D

In this paper, the samples generated by full-wave EM simulation are used to train a 1D-CAE network to obtain the microwave filter responses in a shorter time in the design space. MOEA/D is applied to calculate the geometric parameters of the filter that meet the design requirements. The complete process of microwave filter design by 1D-CAE-network-assisted MOEA/D is given in Figure 4. In this process, there are several key points that need to be described in detail:

(a) Step 2: Data collection step. The purpose of data acquisition is to collect a certain number of training and test sets to develop a surrogate model. A set of data sets consists of a set of geometric parameters of the filter and its corresponding S-parameters. Electromagnetic simulations were performed by a high frequency structure simulator (HFSS) to obtain training and test sets. The values of the geometric parameters ($H$) are determined by the following formulas:

$$H = H_0 + kH_0, \tag{14}$$

where $H_0$ is the initial geometric parameter of the filter (center point), $k$ is a random number in the data collection range, and this range is around the center point and is determined considering the characteristics of the filter in this example.

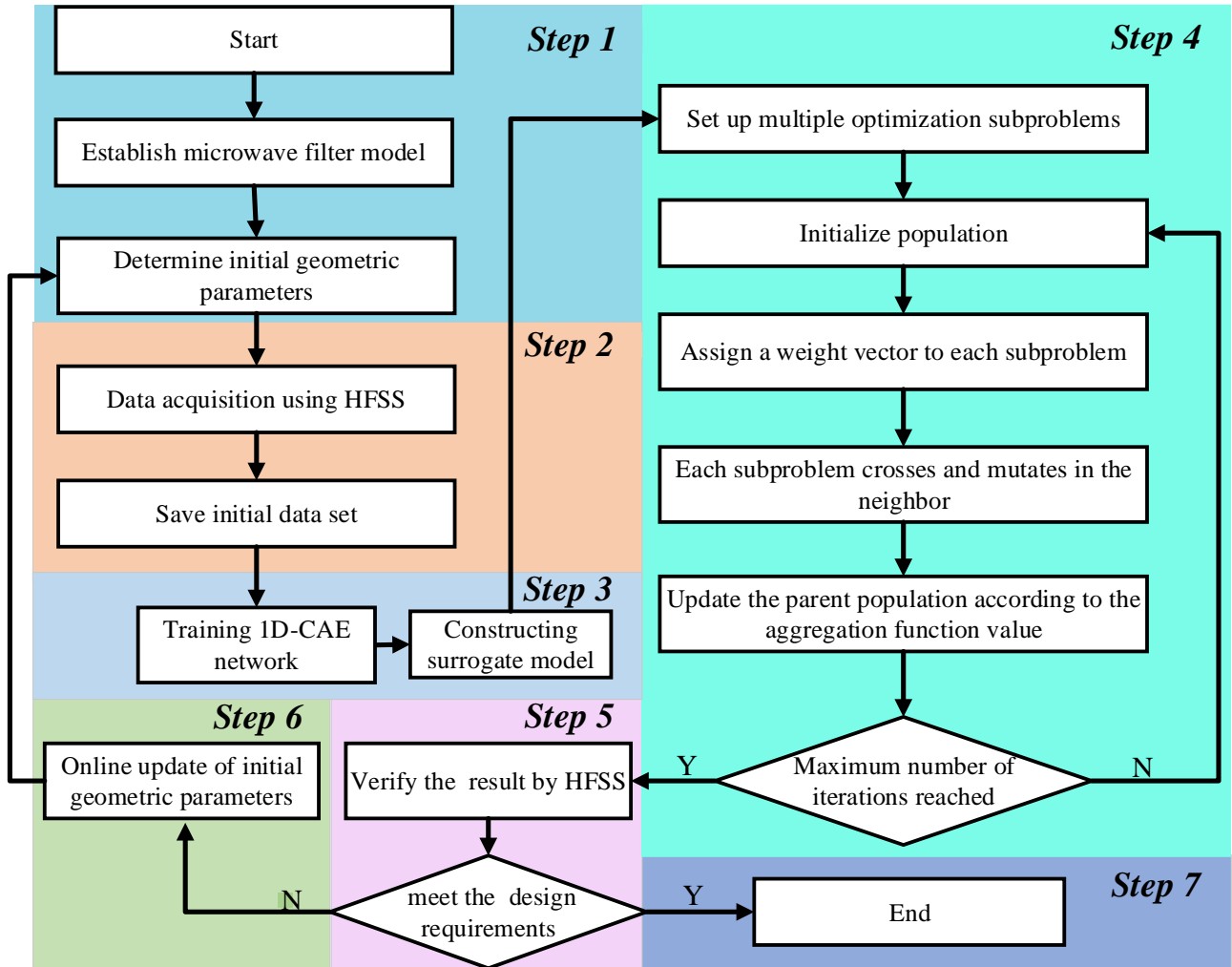

**Figure 4.** The complete flow diagram of the filter design technique.

Next, we input $H_0$, $k$, the number of data acquisitions and the center frequency of the filter, and so on into the data acquisition system. The data acquisition system will automatically drive the HFSS to complete the acquisition process until the preset number of data acquisitions is reached and the system stops running. The entire process requires no human intervention.

(b) Step 3: Surrogate model construction step. In this step, the data set generated in step 2 is divided into test set and training set according to 1:5. The value of the loss function is calculated by Equation (10). The Adam optimizer is used to optimize the loss function of the network.

The complete steps of 1D-CAE-network-aided MOEA/D to design filter are as follows:
**Input:**
⟨1⟩ $H_0$,$k$, the number of data acquisitions;
⟨2⟩ A multi-objective filter design problem with m objectives;
⟨3⟩ Termination conditions (e.g., filter design goal, maximum number of iterations);
⟨4⟩ MOEA/D parameters: the number of sub-problems ($N$); the number of neighbors of each weight vector ($T$); the population size; uniformly distributed weight vector.
**Output:**
⟨1⟩ Filter geometry parameter values that meet design requirements.
*Step 1*: Determine a model step.
  *Step 1.1*: Establish a microwave filter model.
    *Step 1.2*: Determine the initial geometric parameters of the microwave filter, that is, $H_0$ in Equation (14).

*Step 2*: Data collection step.

　*Step 2.1*: The geometric parameter value $H$ is obtained according to Equation (14), and then the response ($S$ parameter) of this microwave filter is generated by HFSS simulation.

　*Step 2.2*: Save the $S$-parameters in the form of the real part and the imaginary part.

*Step 3*: Surrogate model construction step.

　*Step 3.1*: Training 1D-CAE network structure. The value of the loss function is calculated by Equation (10). The Adam optimizer is used to optimize the loss function of the network.

　*Step 3.2*: The surrogate model is constructed via the trained 1D-CAE network structure.

*Step 4:* MOEA/D step.

　*Step 4.1:* Based on the multiple design requirements of the microwave filter, the corresponding optimization subproblems are set up.

　*Step 4.2*: Initialize population.

　　*Step 4.2.1:* Create an external population (EP) to store outstanding individuals, initially empty.

　　*Step 4.2.2:* Calculate the Euclidean distance between any two weight vectors and find the $T$ closest weight vectors to each weight vector. For $i = 1, 2, \ldots, N$, let $B(i) = \{i_1, \ldots, i_T\}$. $x^1, x^2, \ldots, x^N$ are the nearest $T$ weight vectors of $w_j$.

　　*Step 4.2.3:* Generate an initial number of random $x^1, x^2, \ldots, x^N$, using the surrogate model to calculate the fitness function value.

　　*Step 4.2.4:* Initialize $z = \{z_1, \ldots, z_M\}$.

　*Step 4.3*: Assign a weight vector to each subproblem.

　*Step 4.4*: Each subproblem crosses and mutates in the neighbor.

　*Step 4.5*: Update parent population according to the aggregation function value. For $z_j = f_j(y^{\cdot})$,

　　*Step 4.5.1:* Copy: randomly choose two indices *l*, *q* from *B(i)*, and then generate a new solution *y* from $x^l$ and $x^q$.

　　*Step 4.5.2:* Repair: if an element of *y* exceeds a preset bound, its value will be reset to the max or min of the bound (generates $y^{\cdot}$ from *y*).

　　*Step 4.5.3:* Updatez: For each $j = 1, 2, \ldots, m$, judge whether *y* can replace the original extreme value. If $z_j < f_j(y^{\cdot})$, set $F(y^{\cdot})$.

　　*Step 4.5.4:* Update the domain solution *B(i)* for each weight vector $w_j$ in the domain; if it is optimized, update it.

　　*Step 4.5.5:* Update EP: Remove all vectors dominated by $F(y^{\cdot})$ from EP. Add $F(y^{\cdot})$ to EP if no vector dominates $F(y^{\cdot})$ in outer population (EP).

　*Step 4.6*: If the maximum number of iterations is reached, go to step 5. Otherwise, go to Step 4.2.

*Step 5*: Verification optimization result step.

　*Step 5.1*: The optimization results obtained in step 5 are simulated with HFSS.

　*Step 5.2*: If the results meet the design requirements, go to step 7. Otherwise, go to step 6.

*Step 6*: Update the geometric parameters online and then go to step 1.2 directly.

*Step 7*: Complete the design of a microwave filter.

## 3. Design Results

### 3.1. Sixth-Order Ceramic Filter

The first example shows the optimization design of a sixth-order ceramic bandpass filter. The structure of the sixth-order ceramic filter is shown in Figure 5. The finite transmission zeros of this microwave filter are $f_1 = 2.552$ GHz and $f_2 = 2.858$ GHz. The design geometric variables are H = [*h1*, *h2*, *h3*, *h4*, *h5*, *h6*]$^T$, and the design requirements of this filter are as follows:

- $|S11| \leq -20$ dB, for 2.6 GHz $\leq \omega \leq 2.8$ GHz;

- $|S21| \leq -50$ dB, for $\omega$ = 2.552 GHz;
- $|S21| \leq -50$ dB, for $\omega$ = 2.858 GHz.

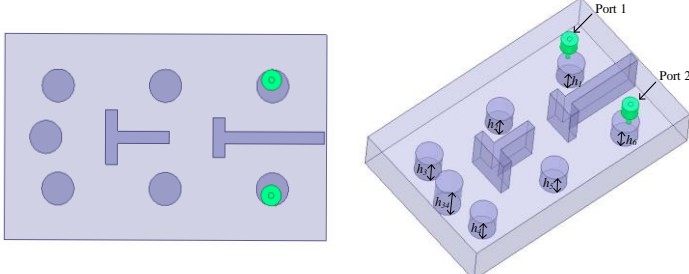

**Figure 5.** Structure of sixth-order ceramic filter.

The starting geometric parameter value of this example is $H = [3.4, 3.3, 3.85, 3.9, 3.4, 3.4]^T$ (mm), and its EM response indicates that the response at the starting point is far away from the design requirements.

This example consists of two main steps. The first step is to develop a surrogate model based on 1D-CAE with the real part and imaginary part of the filter's *S11* and *S21* as the model output and geometrical parameters as model inputs, i.e., $H = [h1, h2, h3, h4, h5, h6]^T$. The EM data are used to train the 1D-CAE. The second step is to apply the trained surrogate model to the multi-objective optimization design, and the geometric parameters can be adjusted repeatedly in the optimization process.

At first, the engineer needs to construct a 1D-CAE network structure. In Figure 6, the encoder part has a convolution layer, pooling layer (down-sampling), and full connection layer. A three-layer full connection constitutes the decoder part. The Conv1D (3, 32, 1) indicates a convolution layer. The first number (3) indicates that the convolution kernel is 3, the second number (32) indicates the number of output channels of this layer is 32 and the third number (1) indicates the stride is 1. Maxpool (3,3) represents a maximum pooling layer with a down-sampling factor value of 3 and stride size of 3. Full connection (N) represents a full connection layer with N nodes, where N is the dimension of the geometric parameters to be predicted by the filter. The input information includes real and imaginary parts of the *S*-parameters of 301 frequency points. After Conv1D (3, 32, 1), the input data are transformed into a matrix of 301 × 32 (32 channels, 301 indicates the length of a single channel). Next, the matrix became 101 × 32 after Maxpool (3,3).

Developing a 1D-CAE network structure requires a certain number of training sets and test sets. EM simulations are performed by a high-frequency structure simulator (HFSS) to obtain training sets and test sets. We set the range of the surrogate model to be $k = [2\% \ 2\% \ 2\% \ 2\% \ 2\% \ 2\%]^T$. This range is around the center point and is determined considering the characteristics of the filter in this example. Every time the geometric parameters change, the filter model is simulated based on this through the EM solver to gain the real part and imaginary part sample data of *S11* and *S21*. So as to ensure the high accuracy of this model, 360 sets of data are obtained, including 288 training sample data and 72 test sample data. It should be noted that in the design process of the microwave filter, tuning is usually the most time consuming and may take several months. In contrast, the efficiency of this approach is sufficient even without parallel computing. If parallel computing is applied, the efficiency is quite high. In Figure 7, the comparison between EM responses and 1D-CAE network structure prediction is shown. It can be observed that the two results are completely consistent. Table 1 shows a comparison of the two approaches (EM responses and 1D-CAE network structure prediction) in terms of CPU time. The EM responses time is viewed through the HFSS simulation process information interface after the simulation. 1D-CAE network structure prediction time is calculated by Equation (15). It can be observed

that 1D-CAE not only achieves almost the same accuracy as HFSS simulation but also completes faster than EM simulation.

$$time_{1D-CAE} = \frac{time\_total}{n},\qquad(15)$$

where *time_total* is the total predicted time of 1D-CAE. *n* is the number of 1D-CAE predictions over this total time.

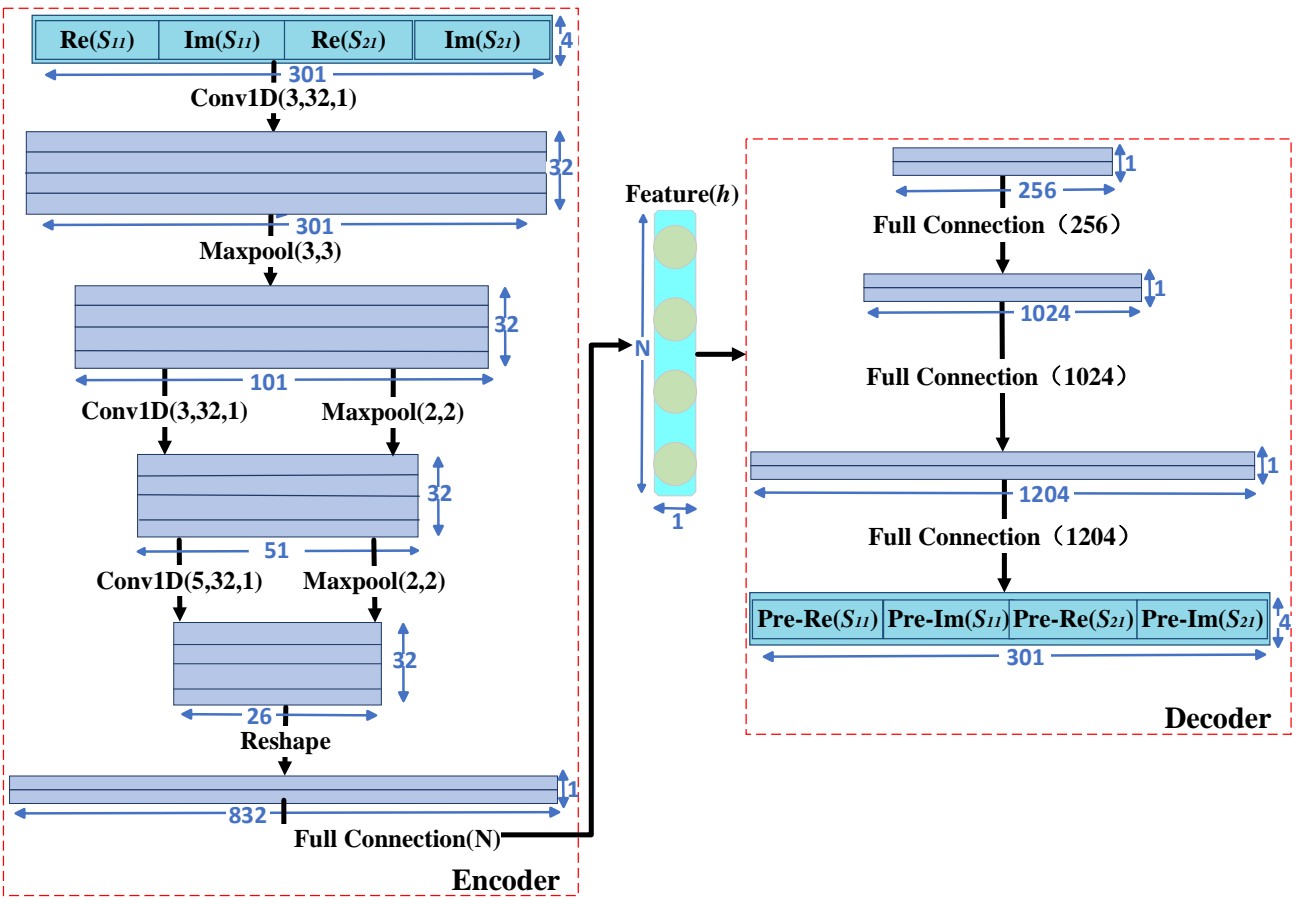

**Figure 6.** Detailed parameters of 1D−CAE network structure.

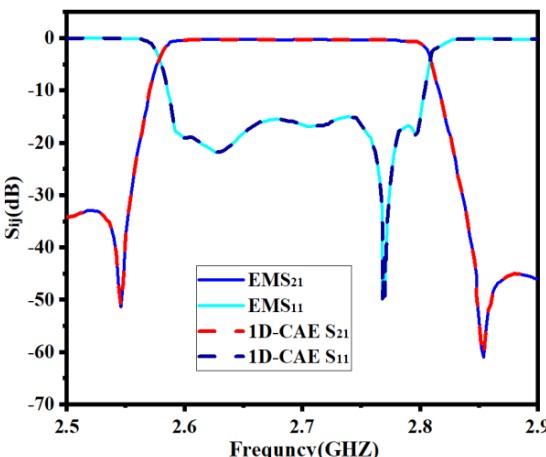

**Figure 7.** Comparison between full−wave EM simulation result and 1D−CAE prediction result of the first example.

**Table 1.** Comparison of CPU time between HFSS and 1D-CAE.

|  | EM Simulation | 1D-CAE Prediction |
|---|---|---|
| Completion time | 3 min | 0.017 s |

Once the surrogate model is trained well, it can be applied to the MOEA/D, in which the geometric parameters can be adjusted repeatedly. In MOEA/D, multiple design objectives of the filter are set as multiple subproblems. A low-complexity surrogate model based on 1D-CAE is constructed to form the results of the MOEA/D scalar subproblem, which tremendously improves the design efficiency. According to the design requirements of the filter, three objective functions are set for optimization:

$$Fit_1 = \max\{(f_l \leq db(S_{11}) \leq f_h)\} - (-20), \tag{16}$$

$$Fit_2 = \min\{(f_1 - 0.05 \leq db(S_{21}) \leq f_1 + 0.05)\} - (-50), \tag{17}$$

$$Fit_3 = \min\{(f_2 - 0.05 \leq db(S_{21}) \leq f_2 + 0.05)\} - (-50), \tag{18}$$

where $f_l$ is 2.6 GHz and $f_h$ is 2.8 GHz. $f_1$ = 2.552 GHz and $f_2$ = 2.858 GHz are the finite transmission zeros of this sixth-order ceramic bandpass filter.

About MOEA/D, the number of neighbors of each weight vector is set to 5, the population size is set to 350, and the maximum number of iterations is set to 60. In order to clearly show the optimization trend, every 10 iterations are defined as a stage, and the optimization results are extracted once. In the sixth stage, MOEA/D found the optimal solution of this sixth-order ceramic bandpass filter. Detailed general flow of this bandpass filter design is shown in Figure 8. The geometric parameters change in the whole process is shown in Table 2. The EM responses at different stages are shown in Figure 9.

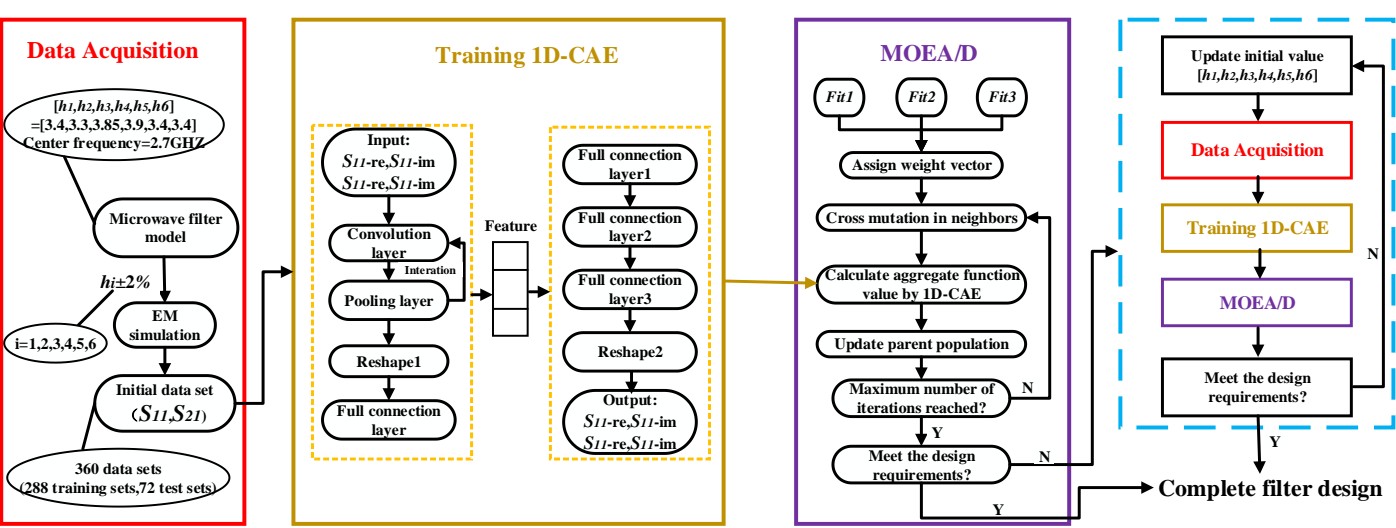

**Figure 8.** General design flow of the filter.

**Table 2.** The geometric parameters' change in the whole process.

| Stage | h1 | h2 | h3 | h4 | h5 | h6 |
|---|---|---|---|---|---|---|
| 0 | 3.400 | 3.300 | 3.850 | 3.900 | 3.400 | 3.400 |
| 1 | 3.374 | 3.365 | 3.845 | 3.883 | 3.370 | 3.407 |
| 2 | 3.350 | 3.370 | 3.845 | 3.853 | 3.370 | 3.385 |
| 3 | 3.335 | 3.370 | 3.845 | 3.845 | 3.370 | 3.360 |
| 4 | 3.330 | 3.370 | 3.845 | 3.845 | 3.370 | 3.357 |
| 5 | 3.330 | 3.369 | 3.845 | 3.845 | 3.370 | 3.348 |
| 6 | 3.329 | 3.369 | 3.845 | 3.845 | 3.373 | 3.330 |

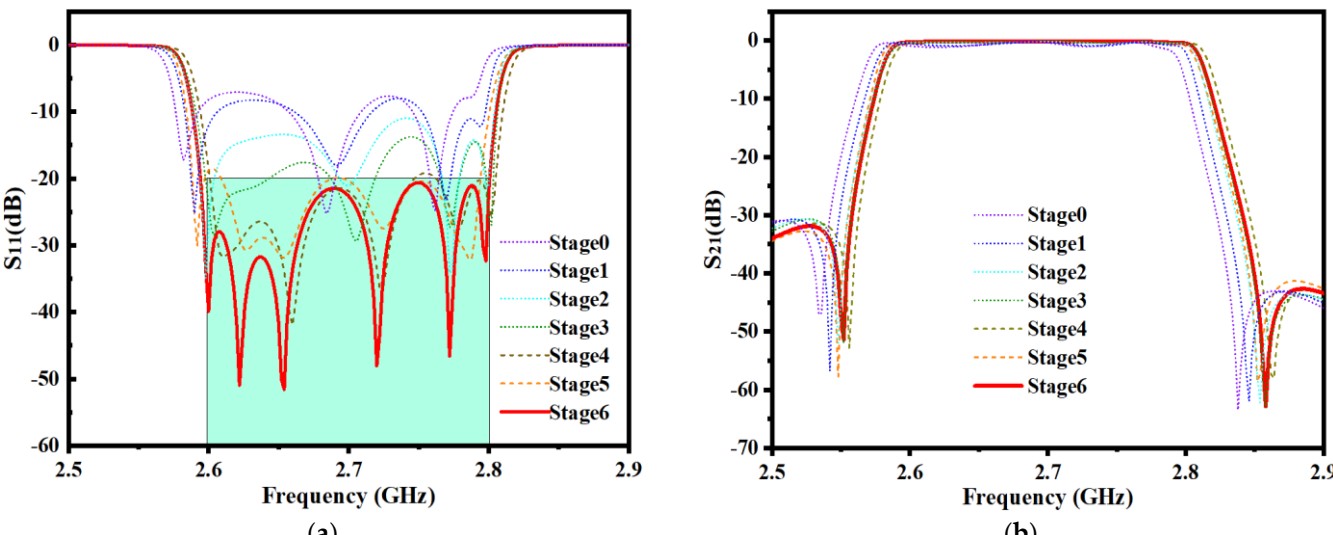

**Figure 9.** EM responses at different steps of the first example. (**a**) Return loss *S11*. (**b**) Insertion loss *S21*.

The comparisons of the proposed optimization and directly using the EM solver for MOEA/D are shown in Table 3. (The data values in Table 3 are missing because using the EM solver directly does not require training a surrogate model.) We can see that our proposed method can achieve an optimal EM solution in less time.

**Table 3.** Comparison of CPU time between two optimization design methods.

|  | Directly Using EM Simulation for MOEA/D | Proposed Optimization |
|---|---|---|
| Total EM simulation time | 184.5 h | 22.5 h |
| Time of surrogate model training | – | 5 min |
| MOEA/D optimization time | 185h | 3.5min |
| Total time | 185h | 22.64h |

### 3.2. Seventh-Order Metal Cavity Bandpass Filter

In the second example, let us consider a seventh-order metal cavity bandpass filter with a center frequency of 1.791 GHz and a bandwidth of 30 MHz, as shown in Figure 10. The finite transmission zero of this seventh-order metal cavity bandpass filter is $f_1 = 1.765$ GHz. The design geometric variables are $H = [w_{12}/w_{67}, w_{23}/w_{56}, w_{34}/w_{45}, h_1/h_7, h_2/h_6, h_3/h_5, h_4]^T$. The design requirements of the bandpass filter are as follows:

- $|S11| \leq -20$dB, for 1.776 GHz $\leq \omega \leq$ 1.806 GHz;
- $|S21| \leq -90$dB, for $\omega = 1.765$ GHz;
- $|S21| \leq -40$dB, for $\omega = 1.825$ GHz.

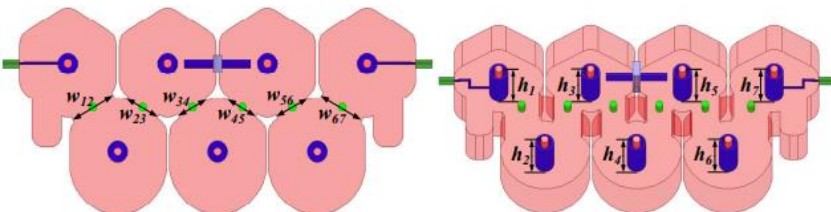

**Figure 10.** Structure of seventh-order metal cavity bandpass filter.

The starting geometric parameter value of this example is H = [22.2, 17.23, 15.7, 6.63, 5.36, 5.6, 5.7]$^T$ (mm), and its EM response indicates that the response at the starting point is far away from the design requirements.

The parameter values of each layer of 1D-CAE network structure are consistent with Figure 6. So as to guarantee the high accuracy of the 1D-CAE network structure, 550 sets of data, including 440 training sets and 110 test sets, are obtained by HFSS. In Figure 11, the comparison of EM responses with 1D-CAE network structure prediction is shown. Table 4 shows a comparison for the two approaches (EM responses and 1D-CAE network structure prediction) in terms of the CPU time.

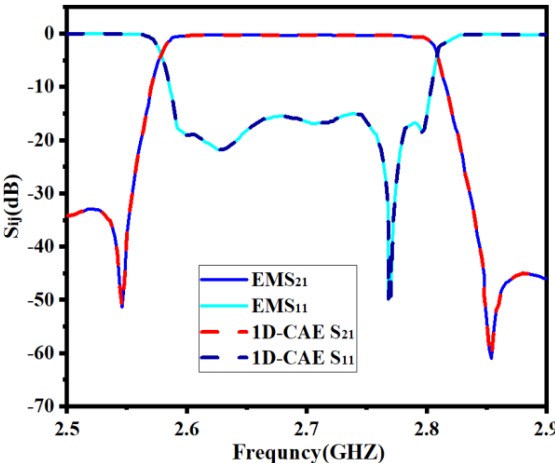

**Figure 11.** Comparison between full−wave EM simulation result and 1D−CAE prediction result of the second example.

**Table 4.** Comparison of CPU time between HFSS and 1D-CAE.

|  | **EM Simulation** | **1D-CAE Prediction** |
|---|---|---|
| Completion time | 12 min | 0.02 s |

According to the design requirements of this filter, three objective functions are set for optimization:

$$Fit_1 = \max\{(f_l \le db(S_{11}) \le f_h)\} - (-20), \tag{19}$$

$$Fit_2 = \min\{(f_1 - 0.05 \le db(S_{21}) \le f_1 + 0.05)\} - (-90), \tag{20}$$

$$Fit_3 = \min\{(f_2 - 0.05 \le db(S_{21}) \le f_2 + 0.05)\} - (-40), \tag{21}$$

where $f_l$ is 1.776 GHz, $f_h$ is 1.806 GHz, $f_1$ = 1.765 GHz, and $f_2$ = 1.825 GHz.

MOEA/D parameters are the same as in the first example, but the maximum number of iterations is set to 50. In order to clearly show the optimization trend, every 10 iterations are defined as a stage, and the optimization results are extracted once. In the fifth stage, MOEA/D found the optimal solution. The geometric parameters' change in the iterative process is shown in Table 5. The EM responses at different stages are shown in Figure 12.

**Table 5.** The geometric parameters' change in the whole process.

| Stage | $w_{12}/w_{67}$ | $w_{23}/w_{56}$ | $w_{34}/w_{45}$ | $h_1/h_7$ | $h_2/h_6$ | $h_3/h_5$ | $h_4$ |
|---|---|---|---|---|---|---|---|
| 0 | 22.200 | 17.230 | 15.700 | 6.630 | 5.360 | 5.600 | 5.700 |
| 1 | 22.192 | 17.233 | 15.679 | 6.586 | 5.398 | 5.657 | 5.720 |
| 2 | 22.507 | 17.155 | 15.684 | 6.586 | 5.398 | 5.660 | 5.720 |
| 3 | 22.505 | 17.153 | 15.689 | 6.581 | 5.397 | 5.643 | 5.723 |
| 4 | 22.500 | 17.158 | 15.690 | 6.581 | 5.401 | 5.645 | 5.721 |
| 5 | 22.443 | 17.154 | 15.689 | 6.580 | 5.401 | 5.645 | 5.721 |

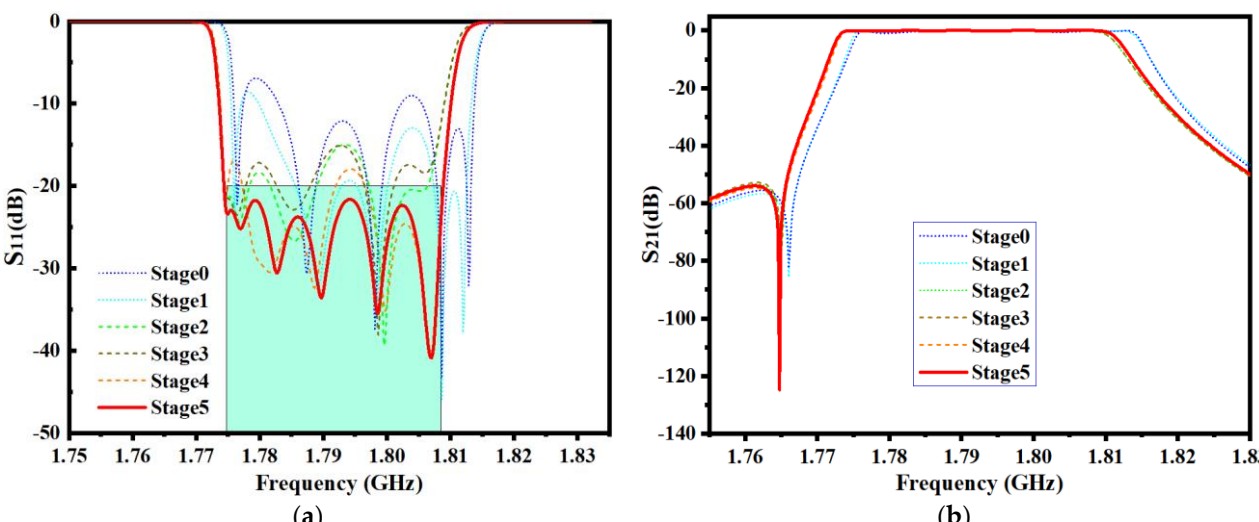

**Figure 12.** EM responses at different steps of the second example. (**a**) Return loss *S11*. (**b**) Insertion loss *S21*.

The comparisons of the proposed optimization and directly using the EM solver for MOEA/D are shown in Table 6. (The data values in Table 6 are missing because using the EM solver directly does not require training a surrogate model.) We can see that our proposed method can achieve an optimal EM solution in less time.

**Table 6.** Comparison of CPU time between two optimization design methods.

|  | Directly Using EM Simulation for MOEA/D | Proposed Optimization |
|---|---|---|
| Total EM simulation time | 615.5 h | 110 h |
| Time of surrogate model training | – | 7 min |
| MOEA/D optimization time | 616 h | 5 min |
| Total time | 616 h | 110.2 h |

## 4. Conclusions

A novel method merging MOEA/D and a 1D-CAE network structure is proposed to design microwave filters. In this new design method, the 1D-CAE network structure is introduced to replace the traditional full-wave simulation, which greatly reduces the design time of the filter. MOEA/D disassembles the MOPs into a set of single-objective optimization subproblems with neighborhood relations, and each subproblem is a dissimilar set of all objectives. Then, by analyzing the information of adjacent problems, all of the subproblems are optimized by an evolutionary algorithm concurrently. This new method is used to design two microwave filters in our examples. The design results show that the new method merging MOEA/D and the 1D-CAE network structure can be used to design filters that meet the requirements easily. With the help of the 1D-CAE network structure, the MOEA/D algorithm more easily avoids falling into the local optimum and completes the filter design in a shorter time than the EM-based microwave filter design.

In future work, we will continue our research from the following two aspects. First, we will try to apply the proposed technique to the design of other passive devices, such as antennas, couplers, duplexers, etc. Second, we will seek more reliable optimization methods to improve the generalization ability of surrogate models.



**Author Contributions:** Conceptualization, Y.W. (Yongfeng Wei), Y.Z., G.Q., and Yanxing Wang; methodology, Y.W. (Yongfeng Wei), Y.Z., G.Q., and Y.W. (Yanxing Wang); software, Y.W. (Yongfeng Wei), Y.Z., G.Q., and N.Y.; validation, Y.W. (Yongfeng Wei), Y.Z., G.Q., and N.Y.; formal analysis, Y.W. (Yongfeng Wei), Y.Z., and L.F.; investigation, N.Y.; resources, Y.W. (Yongfeng Wei), Y.Z., G.Q., and Y.W. (Yanxing Wang); data curation, Y.W. (Yongfeng Wei), Y.Z., G.Q.; writing—original draft preparation, Y.W. (Yongfeng Wei), Y.Z., and G.Q.; writing—review and editing, Y.W. (Yongfeng Wei), Y.Z., and L.F.; visualization, Y.W. (Yongfeng Wei), Y.Z., and G.Q.; supervision, Y.W. (Yongfeng Wei) and Y.Z.; project administration, Y.W. (Yongfeng Wei) and Y.Z.; funding acquisition, Y.W. (Yongfeng Wei) and Y.Z. All authors have read and agreed to the published version of the manuscript.

**Funding:** This work was funded by the National Natural Science Foundation of China (NSFC) under project no. 61761032 and no. 62161032 and the Nature Science Foundation of Inner Mongolia under contract no. 2019MS06006. This work was funded by the Inner Mongolia Foundation 2020MS05059 and Inner Mongolia Department of Transportation NJ-2017-8. This work was also supported by the Shaanxi Key Laboratory of Deep Space Exploration Intelligent Information Technology under grant no. 2021SYS-04. This work was also supported by the Research and Development of New Energy Vehicle Product Testing Conditions in China—Hohhot Baotou, 2017.

**Institutional Review Board Statement:** Not applicable.

**Informed Consent Statement:** Not applicable.

**Data Availability Statement:** Not applicable.

**Conflicts of Interest:** The authors declare no conflict of interest.

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
