# Peer review of "Efficient Microwave Filter Design by a Surrogate-Model-Assisted Decomposition-Based Multi-Objective Evolutionary Algorithm"

_electronics, doi:10.3390/electronics11203309_

Round 1

Reviewer 1 Report

The paper provides a surrogate model based on one-dimensional convolutional autoencoders (1D-CAE) into the multi-objective algorithm evolutionary based on decomposition (MOEA/D) for the first time. Although, tThe authors well presented the paper, there are some points to consider in the revised versions as follows:

1- In the introduction part, it is advisable if possible to reorganize this section into three subsections; (1) Background and motivation, (2) Paper contribution, and (3) Paper organization.

2- In Section 2.2.2., Please use numbering style that do not be confusing, you may use I, II, ... or other subnumbering style.

3- Please use the recommended reference style of the journal where most references have different styles such as [3]-[24].

Reviewer 2 Report

Recommendations

In this article, “Efficient Microwave Filter Design by Surrogate Model Assisted  Decomposition-Based Multi-Objective Evolutionary Algorithm. This manuscript proposed the applies a novel surrogate model based on one-dimensional convolutional autoencoders (1D-CAE) into the multi-objective algorithm evolutionary based on decomposition (MOEA/D) for the first time. The simulation results verified that the surrogate model based on 1D-CAE is used to generate the results of scalar subproblems of MOEA/D, which greatly improves the design efficiency. The topic is interesting and timely. Please, consider the following points:

1-       The authors should explain the importance of using artificial neural networks in terms of the input layer, hidden layer and output layer with activation functions and make them more related to (the encoder part and decoder part).

2-       How many hidden layers and corresponding nodes are required should be elaborated to improve computational efficiency.

3-       The authors should support the equations with citation references from Eq.6 to Eq. 10.

4-       To make the complete process of microwave filter design by 1D-CAE in figure 4 more interactive, must make the stages more related and sequential to the equations.

5-       The Materials and Methods for this article are not enough, THE AUTHORS MUST support technically this section according to the contributions in the abstract.

6-       Please add the equation for EM responses and 1D-CAE network structure prediction) in terms of CPU time in Materials and Methods.

Reviewer 3 Report

Comments to the Author  

Manuscript: Efficient Microwave Filter Design by Surrogate Model Assisted Decomposition-Based Multi-Objective Evolutionary Algorithm (electronics-1940497)  

In this manuscript, the authors explored an Efficient Microwave Filter Design by Surrogate Model Assisted Decomposition-Based Multi-Objective Evolutionary Algorithm uses 1D-CAE to generate the results of scalar  subproblems of MOEA/D, which greatly improves the design efficiency.

Overall, the studied  topic is timely, and the presentation follows a logical style.  

However, the major weakness of the manuscript is its limited contribution.

Besides, this reviewer has also some major concerns regarding the assumptions detailed below.

1. The authors should enrich the literature review section by reviewing more related research on 1D-CAE  and MOEA/D which are called "Related Works".

2. Start and end states are missed in "Figure 4. The complete flow diagram of the filter design technique". 

3. In Table 6, the value of data is missed or explain why data value is not applicable?

4. Mention the future work in the "Conclusion section".  

5. In this manuscript, a lot of mathematical equations are used but do not mention references. Are all equations derived by yourself? 

Round 2

Reviewer 1 Report

The paper is accepted.

Reviewer 2 Report

In this article, “Efficient Microwave Filter Design by Surrogate Model Assisted Decomposition-Based Multi-Objective Evolutionary Algorithm. The authors improve the manuscript according to the new comments in the second revision.

 Accept 

Reviewer 3 Report

Dear Authors,

Thank you for addressing my all comments.

Best regards

Md Sipon Miah